# Keratin Expression in Podocytopathies, ANCA-Associated Vasculitis and IgA Nephropathy

**DOI:** 10.3390/ijms25031805

**Published:** 2024-02-02

**Authors:** Paraskevi Pavlakou, Harikleia Gakiopoulou, Sonja Djudjaj, Kostas Palamaris, Maria Stella Trivyza, Kostas Stylianou, Dimitrios S. Goumenos, Evangelos Papachristou, Marios Papasotiriou

**Affiliations:** 1Department of Nephrology and Kidney Transplantation, University Hospital of Patras, 26504 Patras, Greece; pavlakoup@upnet.gr (P.P.); m.trivyza@upatras.gr (M.S.T.); dgoumenos@upatras.gr (D.S.G.); epapachr@upatras.gr (E.P.); 21st Department of Pathology, National and Kapodistrian University of Athens Medical School, 34400 Athens, Greece; 3Institute of Pathology, RWTH University of Aachen, 52074 Aachen, Germany; 4Department of Nephrology, University Hospital of Heraklion, 71500 Heraklion, Greece; kstylianu@gmail.com

**Keywords:** keratins, glomerulonephritis, biomarker, end-stage kidney disease

## Abstract

Keratins are the main components of the cell cytoskeleton of epithelial cells. Epithelial cells under stressful stimuli react by modifying their keratin expression pattern. Glomerular diseases are pathological conditions that may lead to loss of kidney function if not timely diagnosed and treated properly. This study aims to examine glomerular and tubular keratin expression in podocytopathies, ANCA-associated vasculitis, and IgA nephropathy and how this expression correlates to clinical outcomes. We included 45 patients with podocytopathies (minimal change disease and focal segmental glomerulosclerosis), ANCA-associated vasculitis, and IgA nephropathy, with or without crescentic lesions, and healthy controls. All tissues were assessed by photon microscopy and immunohistochemistry. Biopsy sections were examined for keratins 7, 8, 18, and 19 expression in the glomerular and tubulointerstitial areas separately. Moreover, we examined how keratin expression was correlated with long-term kidney function outcomes. All four studied keratins had significantly increased glomerular expression in patients with ANCA vasculitis compared to controls and MCD patients. Tubular expression of keratins 7, 8, and 19 was related to kidney outcome in all groups. Patients with crescents had higher expression of all keratins in both glomeruli and tubulointerstitium. The presence of tubular atrophy, interstitial fibrosis, mesangial hyperplasia, and interstitial inflammation did not affect keratin expression. Keratins, an abundant component of renal epithelial cells, have the potential to be featured as a biomarker for kidney function prognosis in patients with glomerular diseases.

## 1. Introduction

Glomerular diseases represent a complex and heterogeneous group of diseases with a wide range of pathological characteristics, which derive from their distinct pathogenetic routes. Their profound heterogeneity means that they remain a major challenge in terms of establishing a certain diagnosis and, consequently, in applying the optimal treatment to prevent long-term irreversible renal damage. Their clinical manifestations converge to common findings of glomerular filtration barrier impairment and tubular reabsorption defects, such as proteinuria, hematuria, and increased serum creatinine. Moreover, additional serum and urine biomarkers have been recently identified and are currently used in practice, facilitating the diagnosis and enabling the prediction of relapses [1,2,3]. Furthermore, genetic testing revealing specific abnormalities linked to glomerulopathies can both help in diagnosis confirmation and reveal familial predisposition [2].

However, the diagnostic gold-standard for glomerular diseases is kidney biopsy, which enables the synthesis of both morphological alterations and the immunofluorescent/immunohistochemical (IF/IHC) profile of renal parenchyma, as well as the findings of electron microscopy examination [4]. Pathologic reports of kidney biopsies should analyze distinctly the alterations of different parenchymal compartments, glomeruli, and tubulointerstitial and report both active (necrosis/crescents, endocapillary hypercellularity, acute tubular injury) and chronic lesions (segmental/global sclerosis, tubular atrophy, interstitial fibrosis). Thus, besides the identification of a precise pathological entity, kidney biopsy also reveals the activity and/or chronicity of the underlying pathological process, framing the potential for damage reversibility after proper treatment [5]. In many cases, the evaluation of a kidney biopsy can prove to be a very tricky process, as a possible overestimation can turn a pattern of injury into an established glomerular disease [6] and lead to a sequelae of unsuccessful therapeutic actions. In that direction, the enforcement of kidney biopsy examination with the emergence of tissue biomarkers will improve not only the diagnostic accuracy but also the prognostic value of it.

While the different categories of glomerulopathies are causally associated with heterogeneous stimuli and diverse pathogenetic routes, they converge towards a common feature, which is the impairment of podocytes structural and/or functional integrity. A compensatory mechanism activated by the kidney in order to cope with podocyte injury and the impeding of the glomerular filtration barrier is the mobilization of parietal epithelial cells (PECs), which share common developmental ancestries and evince a close lineage relationship with podocytes [7,8,9,10,11]. PECs make up the inconspicuous inside lining of the Bowman capsule, being in continuity with the podocytes at the vascular pole and the tubular epithelial cells at the tubular pole of the glomeruli. They also seem to be equipped with fate plasticity, as they have demonstrated a context-dependent capacity to serve as renal progenitors, capable of regenerating podocytes [7,12]. The process of their trans-differentiation towards a different cellular identity seems to be step-wise, proceeding via intermediate states of a phenotypic continuum, characterized by a synchronous expression of both podocytic and PEC markers [13,14]. Moreover, it seems that as an adaptive response to specific pathogenic triggers imposed upon podocytes, PECs can also re-enter the cell cycle and form cellular masses that occupy the Bowman capsule, as observed in the collapsing variant of FSGS, or crescentic glomerulonephritis [7,8,9,10,11]. In such cases, they maintain their PEC identity without acquiring features or expressing markers indicative of podocytic differentiation.

Keratins (K) represent abundant cytoplasmic proteins that form the intermediate filament (IF) network, and their expression is considered a distinguishing characteristic of epithelial cells. Keratins are expressed in a specific manner in different types of organs or epithelial cells and are obligatory heteropolymers, i.e., mature filaments have at least 1 type I and 1 type II keratin in a 1:1 stoichiometry [15]. They have been shown to play a crucial role in cellular responses to different types of stress (i.e., mechanical) or injury to protect the cytoskeleton. Keratin function can be summarized in cell shape preservation as well as mechanical stability conservation and intracellular organization and transport [16]. Furthermore, apart from the aforementioned functional roles, keratins can act as stress proteins, and as such, they are upregulated in various human and experimental disease states concerning the kidneys, liver, and pancreas [17,18,19,20]. More than 50 keratins have been recognized, and although their structure has been decoded, there is still a lot to be elucidated about their functional value through posttranslational modifications and participation in diseases [21]. As keratin expression is considered a distinguishing feature of epithelial differentiation, in renal parenchyma, keratins are detected in both PECs and tubular epithelium. In this study, we performed an immunohistochemical analysis of a keratin panel in a cohort of kidney biopsies derived from patients with heterogeneous glomerulopathies and attempted to correlate their expression patterns with different clinicopathological parameters.

## 2. Results

Our study included 45 patients (27 males and 22 females), with a mean age of 47.95 ± 19.85 years, a mean serum creatinine of 2.35 ± 1.9 mg/dL, a mean proteinuria of 3.76 ± 4.67 g/24 h, and a mean follow-up time of 3.76 ± 2 years, who had been diagnosed with a primary glomerular disease via percutaneous kidney biopsy.

Kidney tissue pathology examination for commonly assessed characteristics showed that more than half of patient samples (54.5%) had significant interstitial fibrosis of more than 26% of the tubulointerstitial area, and vascular hyalinosis was present in 48.8%. The majority of patients had tubular atrophy (68.9%) and glomerulosclerosis (73.3%). Crescents were found in 24 tissue samples (53.3%).

### 2.1. Glomerular Expression of Keratins

Immunohistochemical analysis revealed that in glomeruli, expression of all four examined keratins was observed at the parietal epithelial cells in the Bowman’s capsule, with no expression in cells of the glomerular tufts as shown in previous studies [20]. Glomeruli from patients with crescentic ANCA-associated vasculitis or IgAN showed increased keratin positivity of parietal epithelial cells, especially in cells forming the crescents (Figure 1).

Glomerular expression of all examined keratins was not significantly increased in patients with podocytopathies in comparison to controls. In patients with crescentic ANCA-associated vasculitis, or IgAN, glomerular expression of keratins was more than 20- and 10-fold higher in comparison to controls, respectively. In particular, the glomerular expression of all four studied keratins was found to be significantly increased in patients with ANCA-associated vasculitis compared to controls and patients with MCD (Figure 1). Furthermore, patients with ANCA-associated vasculitis had higher glomerular expression of K7, K18, and K19 compared to patients with FSGS (FSGS vs. ANCA K7: −2.512, 95% CI −4.385 to −0.6385, *p* = 0.0041, K18: −2.599, 95% CI −4.449 to −0.7486, *p* = 0.0024, K19: −3.217, 95% CI −5.357 to −1.076, *p* = 0.0011, respectively). Comparisons between patients with ANCA vasculitis and IgAN showed that only K18 glomerular expression was higher in patients with ANCA in comparison to patients with IgAN (2.421, 95% CI 0.7969 to 4.045, *p* = 0.0012).

### 2.2. Tubulointerstitial Expression of Keratins

In the renal tubules, our findings are comparable with previous studies [20,22]. The expression of K8 and K18, both in control kidney biopsies and in biopsies from patients with glomerulonephritis, was prominent within the epithelial cells, along all tubular segments from proximal tubules to collecting ducts. On the other hand, the expression of K7 and K19 showed a tubular segment-specific pattern that was confined to the collecting ducts. A localization study of keratins has already been shown in a previous study where collecting duct-specific marker aquaporin-2 is co-localized with K7, 8, 18, and 19, while CD13, a marker of proximal tubules, co-localizes with K18 and 8 [20].

Concerning keratin tubular expression among the examined different glomerulopathies, we found overall significant differences only for K7 and K19. Biopsies from patients with podocytopathies (MCD and FSGS) showed no substantial upregulation of tubular keratins. In crescentic ANCA-associated vasculitis and in IgAN, keratin expression was increased 2- to 4-fold. In more detail, K19 tubular expression was found to be significantly higher in patients with ANCA-associated vasculitis compared to controls and patients with MCD and FSGS (−4.278, 95% CI −7.194 to −1.363, *p* = 0.0015, −3.545, 95% CI −6.461 to −0.63, *p* = 0.0108, −3.097, 95% CI −5.71 to −0.48, *p* = 0.0137, respectively). When compared to controls, IgAN patients had higher K19 tubular expression (−3.051, 95% CI −6.011 to −0.089, *p* = 0.041) (Figure 2). Multiple comparison post hoc tests did not detect any other significant upregulation of tubular keratin expression between controls and patients with podocytopathies, ANCA-associated vasculitis, or IgAN.

### 2.3. Keratin Expression in Comparison to Basic Histopathological Features

When we examined the effect of basic pathological kidney biopsy characteristics on keratin expression, we found no significant effect of the degree of IF/TA, mesangial hyperplasia, or interstitial inflammation. Interestingly, the presence or absence of interstitial fibrosis did not affect the expression of keratins in the tubulointerstitial area. Nevertheless, patients with glomerular sclerosis had in general higher tubular expression of all types of keratins (Table 1); however, statistical significance was found only for K8 and K18.

### 2.4. Prognostic Value of Keratin Expression

As derived from patients’ clinical course at the end of the observation period, 55.6% of the studied population presented kidney function deterioration that was defined as either the doubling of baseline serum creatinine or the induction of renal replacement therapy. Patients who showed kidney function deterioration had significantly higher K7, K8, and K19 tubular expression. On the other hand, glomerular keratin expression was not related to the clinical course of the patients (Table 2).

When the clinical course between patients with ANCA vasculitis and IgAN was tested separately, keratin tubular expression was again related to the kidney outcome. Patients with ANCA-associated vasculitis and IgAN who showed deterioration of kidney function had higher tubular expression of K8, K18, and K19 at diagnosis (Table 3). Again, glomerular expression of keratins did not show any differentiation between groups (with or without deterioration).

## 3. Discussion

Our study revealed interesting findings regarding keratin expression, which seems to differ among different types of glomerulopathies. More specifically, crescentic glomerulonephritides were related to a trend for higher expression of examined keratins in both the glomerular and tubulointerstitial compartments. Patients with ANCA-associated vasculitis had significantly increased glomerular expression of keratins, not only in relation to healthy individuals but also when compared with patients with podocytopathies (MCD and FSGS). On the other hand, nephrotic syndrome proteinuria associated with podocytopathies did not cause significant alterations in the keratin expression profile in both glomerular parietal and tubular epithelial cells. Moreover, in cases with noteworthy chronicity lesions (IF/TA) and glomerulosclerosis, no remarkable results were observed regarding keratin expression, with the exception of increased K8 and K18 in the tubulointerstitial area. The most clinically significant finding of our study has to do with the potential predictive value of keratin expression regarding renal outcome, as K7, K8, and K19 were found to have significantly higher tubular expression at diagnosis in patients with doubling of serum creatinine and those reaching end-stage kidney disease post-biopsy during follow-up.

Keratin expression does not follow one specific pattern in all types of tissue and organs. It is well known that keratins frequently undergo post-translational modifications (i.e., glycosylation, caspase cleavage, phosphorylation, etc.), which allow them to re-organize and adapt to surrounding conditions [21]. A number of diseases have been attributed to keratin-inherited mutations [18], while their involvement in tumor pathogenesis has been well demonstrated by multifarious studies. Moreover, they have shown some potential as emerging prognostic biomarkers in some types of cancer [18,23]. Their multidimensional role in both physiology and disease pathogenesis is associated with their ability to directly regulate numerous aspects of cellular biology and homeostasis, including protein synthesis, re-entry in the cell cycle, cell growth control during wound healing, and many more. Regarding renal tissue, normal glomerular epithelial cells and tubular epithelial cells co-express K8 and K18 [23]. It is a well-recognized property of renal parenchymal cells, in analogy with the paradigm set by other tissues, that they are intrinsically capable of modifying the expression patterns of specific keratins as a reaction to a harmful or stressful effect, with de novo induction of K7 and K19 [22], an event that is inversely related to differentiation level [23]. Moreover, this turnover has been linked with malignant transformation and the development of renal tumors in patients with end-stage renal disease [24].

In our study, all four examined keratins had increased glomerular expression in different types of glomerulonephritis with extracapillary or mesangial proliferation, suggesting an increased inflammatory reaction. This expression pattern might serve as a marker of glomerular epithelial cell stress. Furthermore, this is indicative of underlying epithelial cell injury in glomerular diseases and is in accordance with former results where keratin expression was upregulated in experimental mouse models with different types of kidney injury and was maintained during the progression to advanced chronic kidney disease [20].

Patients with ANCA vasculitis had a significantly higher glomerular expression of all four keratins compared to control cases and patients with podocytopathies. This suggests that the extent of keratin upregulation correlates with the extent of glomerular injury. Regarding the tubulointerstitium, patients with ANCA vasculitis had higher expression of K19 only compared to MCD patients. The deviations observed in keratin expression among the two types of glomerulopathies can be imputed to the discrete pathophysiological mechanisms governing their initiation and progression. The severe inflammatory reaction, imprinted in renal biopsy as extracapillary proliferation, crescent formation, and interstitial inflammation [25], which drive the pathogenesis of ANCA-associated glomerulopathy, elicits substantial damage to kidney functional units, which translates mainly to severe podocyte injury as well as tubular cell loss. As a result, the adjacent surviving cells need a major reconstruction of their cytoskeleton, especially their intermediate filament networks, in order to evade the harmful effects of inflammation and survive the stressful conditions imposed upon them. This intermediate filament remodeling is achieved primarily by altering the expression of keratins. On the other hand, in MCD disease, the initial triggers that interfere with glomerular function and prompt podocytic damage are much more subtle, with minimal inflammatory elements participating. Consequently, minimal adaptive mechanisms, including keratin modifications, are required in order for the cells to adapt to glomerular injury. This is further supported by our findings that all patients with crescents present in kidney biopsy had a trend with higher expression of keratins not only in glomeruli but also in the tubulointerstitial area. Possibly the additive impact of tubular atrophy, mesangial hyperplasia, interstitial inflammation, and fibrosis in RPGN may form this observation and explain the fact that our results did not reach the statistical significance threshold independently.

The central finding of our study is the relationship between kidney function deterioration and increased keratin expression at diagnosis. In more detail, tubular expression of K7, K8, and K19 was found to be significantly increased in patients who showed a doubling of serum creatinine, or ESKD, during follow-up. In a sub-group analysis of patients with crescentic glomerulopathies (i.e., ANCA and IgAN), we found a statistically significant increase in tubular K8, K18, and K19 expression in those with kidney function deterioration. There were no major findings regarding glomerular keratin expression or renal outcome in this group of patients. Patients with glomerulosclerosis, an irreversible kidney damage, had higher expression of keratins in general, but statistical significance was found only for K8 and K18 in the tubulointerstitial area. These findings support a probable prognostic value of increased keratin expression in the tubulointerstitial area regarding renal outcome for individuals with kidney injury due to not only ureteral obstruction or acute tubular necrosis [20], but also glomerular diseases. Our finding could indicate that higher tubular epithelial cell stress, as defined by higher tubular keratin expression, could lead to subsequent higher interstitial fibrosis in the long term, which is highly correlated with kidney function deterioration.

A key point in the establishment of chronic kidney disease is the development of interstitial fibrosis and tubular atrophy. There are experimental models that have studied this phenomenon and described an epithelial–mesenchymal transition where tubular epithelial cells undergo phenotypic modifications and differentiate into myofibroblasts that repress keratin expression and acquire the ability to serve as adept de novo collagen producers [26,27,28]. In our study, the presence of interstitial fibrosis did not affect keratin expression. This could be due to the fact that in advanced kidney fibrosis, the number of tubules in the interstitial space is reduced, and therefore the total expression of the quantified staining does not appear to have increased. Nevertheless, the increased keratin expression in the tubulointerstitial area in patients with deterioration of kidney function is challenging former research findings. In line with our results, there are data that not only do not confirm the loss of keratins but support the increased keratin expression when tubular atrophy and interstitial fibrosis are present, indicating chronic kidney disease [20]. This is attributed to the upregulation of single-cell expression of keratins in the tubular epithelial cells.

### Limitations

One of the limitations of our study is that we did not include in our analysis the treatment that patients received, if any, to examine any possible effect on keratin expression and renal outcome. Moreover, although we calculated the intensity of keratin expression, we did not include in our analysis the cellular pattern of it both in the glomeruli and the tubular area. Another limitation is that, due to the retrospective nature of the study, we were not able to conduct measurements of keratin excretion in urine or serum to examine the possible accumulation or excretion element as a possible marker of kidney injury and disease progression. Finally, the inclusion of a larger patient sample may have facilitated the attainment of stronger and even clearer findings.

## 4. Methods and Materials

Patients that were included in the study were comprised of two main groups according to the pathology of kidney biopsy. Those with proven podocytopathies, either minimal change disease (MCD) (N = 7) or focal segmental glomerulosclerosis (FSGS) (N = 8), and those with hyperplastic lesions such as anti-neutrophil cytoplasmic antibodies (ANCA) positive associated vasculitis with rapidly progressive glomerulonephritis with crescents (N = 17) and IgA nephropathy (IgAN) (N = 13). As control biopsies, we used (N = 5) tumor-distant kidney tissue from nephrectomies of patients with renal cell carcinoma without any other kidney disease and normal histopathology. All patients gave their informed consent or were analyzed in a retrospective, anonymized manner. The study protocol is in accordance with the Helsinki Declaration of 1975, as revised in 2013, and has been approved by the Patras University Hospital committee on human research.

### 4.1. Histology, Immunohistochemistry and Immunofluorescence

Renal biopsies were processed for light microscopy after fixation in formaldehyde (paraffin sections cut at 3 μm) and for immunofluorescence microscopy (cryostat sections for IgG, IgA, IgM, C3, C1q, C4, κ and λ light chains, albumin, and fibrinogen). Standard histochemical stains (PAS, Masson trichrome, silver, and Congo red) were performed. PAS and Masson trichrome were used for a more precise evaluation of tubular atrophy and interstitial fibrosis, respectively. Interstitial inflammation, interstitial fibrosis, and tubular atrophy (IF/TA) were evaluated semi-quantitatively in the renal cortex as follows: Mild (10–25%), moderate (26–50%), and severe (>50%) of the renal cortex area, as previously described [29]. For statistical analysis purposes, an IF/TA of 25% or less was regarded as absent and 26% or more as present. Immunohistochemical stains were performed according to an indirect immunoperoxidase protocol as described previously [30]. The primary and secondary antibodies used in this study are listed in Appendix A. Negative controls were assessed as per protocol and consisted of the substitution of all primary antibodies for keratins 7, 8, 18, and 19 with equivalent concentrations of rabbit IgG for keratin 8 and mouse IgG for keratins 7, 18, and 19.

The evaluation of immunohistochemistry was performed on whole slide images scanned by Aperio ImageScope v.12.4 (Leica, Deer Park, IL, USA) and using the ImageJ v1.52a software (http://imagej.nih.gov/ij/). The percentage of positively stained area in each tissue was calculated separately in interstitial fields at 400× magnification, representing the entire biopsy area as described previously [31,32]. Furthermore, the glomerular positively stained area was calculated in each glomerulus separately by freehand selecting the appropriate area in each picture. Kidney biopsy with less than 8 glomeruli was excluded from the study.

### 4.2. Statistical Analyses

All continuous data are presented as means ± standard deviation (SD). The Kolmogorov–Smirnov test was used to examine data distribution normality. Normally distributed data were compared with a simple t-test and skewed data with a Mann–Whitney test. ANOVA or Friedman test with Tukey’s or Dunn’s post-hoc analysis was used to investigate the differences among the means of each keratin immunohistochemical expression, either glomerular or interstitial, between the different groups of patients in normally distributed and skewed data, respectively.

All tests were 2-tailed, and statistical significance was defined for *p*-values < 0.05. All statistical analyses were performed using SPSS for Windows (version 16.0 SPSS Inc., Chicago, IL, USA) and GraphPad Prism (version 5.00 for Windows, GraphPad Software, San Diego, CA, USA).

## 5. Conclusions

The complexity of diagnosing and staging kidney diseases is an open issue for clinicians. Directing research towards the designation of biomarkers with a predictive value that requires less demanding techniques to be studied is optimum. Utilizing tissue-native molecules for these purposes, such as keratins, is an approach with substantial potential. According to the so far available data, keratin expression in renal parenchyma has prognostic characteristics for renal outcomes that remain to be featured in more detail with further studies.

## Figures and Tables

**Figure 1 ijms-25-01805-f001:**
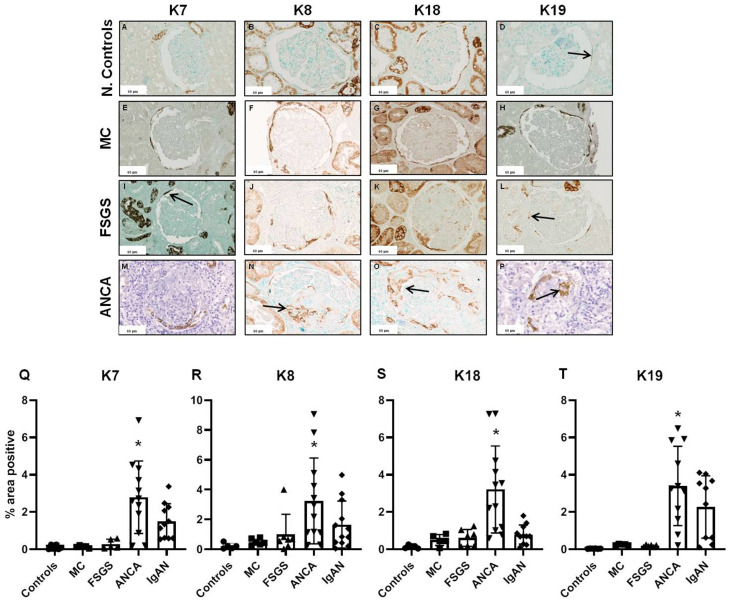
Glomerular keratin expression in normal controls (**A**–**D**), patients with podocytopathies (MCD: (**E**–**H**) and FSGS: (**I**–**L**)) and ANCA vasculitis: (**M**–**P**). Quantification of glomerular expression of keratins is presented in (**Q**–**T**). Scale bars represent 60 μm. (arrows in (**A**,**I**,**L**) show keratins expression in parietal epithelial cells, arrows in (**N**,**O**,**P**) show keratins expression in crescents), (* *p* value < 0.05).

**Figure 2 ijms-25-01805-f002:**
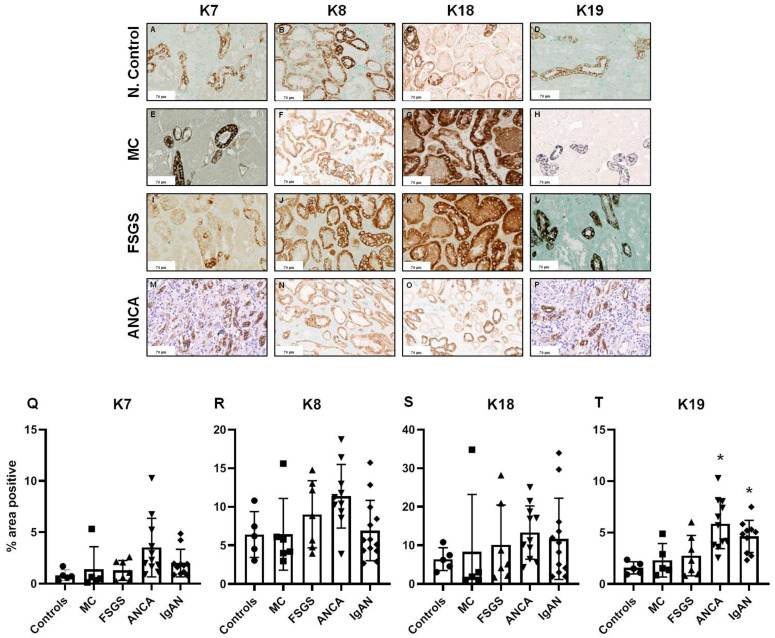
Tubular expression of keratins in normal controls (**A**–**D**) patients with podocytopathies (MCD: (**E**–**H**) and FSGS: (**I**–**L**)) and ANCA vasculitis: (**M**–**P**). Quantification of tubular expression of keratins is presented in (**Q**–**T**). Scale bars represent 70 μm. (* *p* value < 0.05).

**Table 1 ijms-25-01805-t001:** Tubular keratin expression in relation to glomerular sclerosis.

	No Significant Glomerular Sclerosis	Significant Glomerular Sclerosis	*p*-Value
Tubular keratin expression (positive stained area ± SD)			
K7	1.125 ± 0.73	2.31 ± 2.23	0.113
K8	5.131 ± 2.89	9.57 ± 4.59	0.001
K18	4.85 ± 4.29	12.6 ± 9.82	0.003
K19	2.93 ± 1.5	4.32 ± 2.46	0.096

**Table 2 ijms-25-01805-t002:** Keratin expression in relation to patients’ renal outcome.

	Patients with Stable Kidney Function	Patients with Kidney Function Deterioration	*p*-Value
Tubular keratin expression (positive stained area ± SD)			
K7	1.144 ± 0.76	2.755 ± 2.31	0.034
K8	6.489 ± 3.78	11.498 ± 4.11	0.006
K18	9.409 ± 8.44	16.953 ± 11.75	0.081
K19	3.092 ± 1.64	5.24 ± 2.55	0.024
Glomerular keratin expression (positive stained area ± SD)			
K7	1.059 ± 1.05	1.02 ± 1.15	0.945
K8	1.846 ± 2.47	1.143 ± 1.41	0.502
K18	0.899 ± 0.69	0.642 ± 0.37	0.415
K19	0.93 ± 1.38	0.272 ± 0.21	0.171

**Table 3 ijms-25-01805-t003:** Keratin expression in patients with ANCA-associated vasculitis and IgAN with stable or deteriorated kidney function.

	Patients with Stable Kidney Function (Positive Stained Area ± SD)	Patients with Kidney Function Deterioration (Positive Stained Area ± SD)	*p*-Value
Tubular keratin expression			
K7	1.62 ± 0.41	3.137 ± 2.43	0.189
K8	6.676 ± 3.58	12.819 ± 10.37	0.007
K18	9.475 ± 4.66	19.067 ± 10.37	0.045
K19	3.696 ± 1.26	6.275 ± 2.41	0.038
Glomerular keratin expression			
K7	1.754 ± 0.94	1.494 ± 1.14	0.704
K8	3.113 ± 3.25	1.611 ± 1.8	0.438
K18	1.333 ± 0.76	0.738 ± 0.37	0.198
K19	1.625 ± 1.75	0.365 ± 0.37	0.191

## Data Availability

Data of this paper are available at editor’s request.

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
