# Peer review of "Keratin Expression in Podocytopathies, ANCA-Associated Vasculitis and IgA Nephropathy"

_ijms, 2024, doi:10.3390/ijms25031805_

Round 1
Reviewer 1 Report
Comments and Suggestions for Authors
The manuscript performed an immunohistochemical analysis of a keratin panel in a cohort of kidney biopsies, derived from patients with heterogeneous glomerulopathies and attempted to correlate their expression patterns with different clinicopathological parameters. They reported that Keratins, including keratins 7, 8, 18, and 19, an abundant component of renal epithelial cells, have the potential to be featured as a biomarker for kidney function prognosis in patients with glomerular diseases. The reviewer has some concerns that should be addressed.
1. The specificity of antibodies for keratins 7, 8, 18, and 19 should be clarified.
2. Localization of keratins 7, 8, 18, and 19 in the kidney should be determined by co-labeling with specific markers.
3. How about the levels of keratins 7, 8, 18, and 19 in the urine?
Author Response
Thank for your remarks. You can see our reply in the attached file.

Reviewer 2 Report
Comments and Suggestions for Authors
This is a potentially interesting article, although it remains somewhat anecdotal.
1. The term "hyperplastic glomerulopathy" is somewhat strange.
2. It is unclear why the authors choose ANCA-associated vasculitis.
3. As the authors mentioned most biopsy samples showed much chronic lesions. Thus, the expression of keratins may be sequale of chronic inflammation. If so, the implication of this study adding to our current knowledegs not relevant.
4. It is unclear how many glomeruli examined in each study.
5. I do not think the results obtained leading to be featured as a biomarler for kidney function as the authors concluded.
Comments on the Quality of English LanguageMay be OK, bout could be improved.
Author Response
Thank you for your comments. Please see the attached file for our reply.

Round 2
Reviewer 1 Report
Comments and Suggestions for Authors
I have no further comments.
Reviewer 2 Report
Comments and Suggestions for Authors
I understand the authors' responses. I have no further concerns.